

# Atmospheric trace metals measured at a regional background site (Welgegund) in South Africa

**Andrew D. Venter[1], Pieter G. van Zyl[1], Johan P. Beukes[1], Micky Josipovic[1], Johan Hendriks[1], Ville Vakkari[2] and Lauri Laakso[1,2]**

[1]{Unit of Environmental Sciences and Management, North-West University, Potchefstroom, South Africa}

[2]{Finnish Meteorological Institute, Helsinki, Finland}

Correspondence to: P.G. van Zyl (pieter.vanzyl@nwu.ac.za)

## Abstract

Atmospheric trace metals can cause a variety of health-related and environmental problems. Only a few studies on atmospheric trace metal concentrations have been conducted in South Africa. Therefore the aim of this study was to determine trace metals concentrations in aerosols collected at Welgegund, South Africa. $PM_1$, $PM_{1-2.5}$ and $PM_{2.5-10}$ samples were collected for 13 months and 31 atmospheric trace metal species were detected. Atmospheric iron (Fe) had the highest concentrations in all three size fractions, while calcium (Ca) was the second most abundant species. Chromium (Cr) and sodium (Na) concentrations were the third and fourth most abundant species, respectively. The concentrations of the trace metal species in all three size ranges were similar, with the exception of Fe that had higher concentrations in the $PM_1$ size fraction. With the exception of titanium (Ti), aluminium (Al) and manganese (Mg), 70% or more of the trace metal species detected were in the smaller size fractions, which indicated the influence of industrial activities. However, the large influence of wind-blown dust was reflected by 30% and more of trace metals being present in the $PM_{2.5-10}$ size fraction. Comparison of trace metals determined at Welgegund to those in the western Bushveld Igneous Complex indicated that at both locations similar species were observed with Fe being the most abundant. However, concentrations of these trace metal species were significantly higher in the western Bushveld Igneous Complex. Fe concentrations at the Vaal Triangle were similar to levels thereof at Welgegund, while concentrations of species associated pyrometallurgical



smelting were lower. Annual average Ni was four times higher and annual average As was
marginally higher than their respective European standards limit values, which could be
attributed to regional influence of pyrometallurgical industries in the western Bushveld Igneous
Complex. All three size fractions indicated elevated trace metal concentrations coinciding with
the end of the dry season, which could partially be attributed to decreased wet removal and
increases in wind generation of particulates. Principal component factor analysis (PCFA)
revealed four meaningful factors in the $PM_1$ size fraction, i.e. crustal, pyrometallurgical-related
and Au slimes dams. No meaningful factors were determined for the $PM_{1-2.5}$ and $PM_{2.5-10}$ size
fractions, which was attributed to the large influence of wind-blown dust on atmospheric trace
metals determined at Welgegund. Pollution roses confirmed the influence of wind-blown dust
on trace metal concentrations measured at Welgegund, while the impact of industrial activities
was also substantiated.

## 1    Introduction

Atmospheric aerosols are either directly emitted into the atmosphere (primary aerosols) from
natural and/or anthropogenic sources, or are formed through gaseous reactions and gas-to-
particle conversions (secondary aerosols). Aerosols have high temporal and spatial variability,
which increases the need and importance for detailed physical and chemical characterisation
on a regional scale in order to assess the impacts of aerosols (Pöschl, 2005). Particulate matter
(PM) is classified according to its aerodynamic diameter, as $PM_{10}$, $PM_{2.5}$, $PM_1$ and $PM_{0.1}$,
which relates to aerodynamic diameters being smaller than 10, 2.5, 1 and 0.1 μm, respectively.
Larger particulates have shorter lifetimes in the atmosphere compared to smaller particles,
while the impacts of these species are also determined, to a large degree, by their size (Tiwari
et al., 2012, Colbeck et al., 2011). The largest uncertainties in the estimation of direct and
indirect radiative forcing from aerosols are related to the insufficient knowledge of the high
spatial and temporal variability of aerosol concentrations, as well as their microphysical,
chemical and radiative properties (IPCC 2014). Aerosols consist of a large number of organic
and inorganic compounds, of which typical inorganic species include ionic species and trace
metals.
Natural sources of atmospheric trace metals include mineral dust, crustal species, oceans and
biomass burning (wild fires), while major anthropogenic sources are pyrometallurgical





processes, fossil fuel combustion and incineration (Pacyna and Pacyna, 2001). Larger aerosol
particles (>2.5 μm) are usually associated with natural emissions through processes such as
rock weathering and soil erosion (Nriagu et al., 1989). Trace metal species usually associated
with natural emissions include sodium (Na), silicon (Si), magnesium (Mg), aluminium (Al),
potassium (K), calcium (Ca), titanium (Ti), chromium (Cr), manganese (Mn) and iron (Fe)
(Adgate et al., 2007). Arsenic (As), barium (Ba), cadmium (Cd), copper (Cu), nickel (Ni), zinc
(Zn), vanadium (V), molybdenum (Mo), mercury (Hg) and lead (Pb) are mostly related to
anthropogenic activities (Pacyna (1998); Polidori et al., 2009). One of the most significant
sources of anthropogenic trace metal emissions is the industrial smelting of metals. Industrial
pyrometallurgical processes produce the largest emissions of As, Cd, Cu, Ni and Zn (Zahn et
al., 2014). Cr, Ba, Mo, Zn, Pb and Cu are typically associated with motor-vehicle emissions
and oil combustion, while Fe, Pb and Zn are emitted from municipal waste incinerators (Adgate
et al., 2007). However, most of these atmospheric trace metals are emitted through a
combination of different anthropogenic sources (Polidori et al., 2009).
Although trace heavy metals, i.e. metals > Ca, represent a relatively small fraction of
atmospheric aerosols (with the exception of Fe that could contribute a few percent) (Colbeck,
2008), these species can cause a variety of health-related and environmental problems, which
depends on the aerosol composition, extent and time of exposure (Pöschl, 2005). The potential
hazard of several toxic species is well documented as discussed, for instance, by Polidori et al.
(2009), indicating that trace metals such as As, Cd, Co, Cr, Ni, Pb and Se are considered human
and animal carcinogens even in trace amounts (CDC, 2015). It has also been shown that Cu,
Cr and V can generate reactive oxygenated species that can contribute to oxidative DNA
damage (Nel, 2005). Furthermore, trace metals such as Cr, Fe and V have several oxidation
states that can participate in many atmospheric redox reactions (Seigneur & Constantinou,
1995), which can catalyse the generation of reactive oxygenated species (ROS) that have been
associated with direct molecular damage and with the induction of biochemical synthesis
pathways (Rubasinghege et al., 2010). Guidelines for atmospheric levels of many trace metals
are provided by the World Health Organization (WHO) (WHO 2005). In addition, lighter
metals such as Si, Al and K are the most abundant crustal elements (next to oxygen), which
can typically constitute up to 50% of remote continental aerosols. These species are usually
associated with the impacts of aerosols on respiratory diseases and climate.



South Africa has the largest industrialised economy in Africa, with significant mining and
metallurgical activities. South Africa is a well-known source region of atmospheric pollutants,
which is signified by three regions being classified through legislation as air pollution priority
areas, i.e. Vaal Triangle Airshed Priority Area (Government Gazette, 2006), Highveld Priority
Area (Government Gazette, 2007) and Waterberg-Bojanala Priority Area (Government
Gazette, 2012). Air quality outside these priority areas is often adversely affected due to
regional transport and the general climatic conditions, such as low precipitation and poor
atmospheric mixing in winter. Only a few studies on the concentrations of atmospheric trace
metals in South Africa have been conducted (Van Zyl et al., 2014; Kgabi, 2006; Kleynhans,
2008). In addition, these studies were also conducted within two of these priority areas
containing a significant number of large point sources, and regional impacts of atmospheric
trace metals could therefore not be assessed.
In this study, trace metals were determined in three size ranges in aerosol samples collected for
one year at the Welgegund atmospheric measurement station in South Africa. Welgegund is a
comprehensively equipped regional background atmospheric measurement station that is ~100
km downwind of the most important source regions in the interior of South Africa (e.g. Tiitta
et al., 2014). In an effort to determine major sources of trace metals on a regional scale, source
apportionment was also performed by applying principal component factor analysis (PCFA).
**2   Experimental**
**2.1   Site description**
Aerosol sampling was performed at Welgegund ([www.welgegund.org](www.welgegund.org), 26°34'11.23"S,
26°56'21.44"E, 1480 m a.s.l. (above sea level)) in South Africa. As indicated in Figure 1,
Welgegund is situated in the interior of South Africa and is frequently affected by air masses
moving over the most important anthropogenic/industrial source regions in the interior (Beukes
et al., 2013, Tiitta, et al., 2014, Jaars, et al., 2014, Vakkari et al., 2015; Booyens et al., 2015).
Also indicated in Figure 1 are the major industrial point sources, i.e. coal-fired power plants,
petrochemical industries and pyrometallurgical smelters. In Beukes et al. (2013) Tiitta, et al.,
2014 and Jaars, et al., 2014, reasons for the site selection, prevailing biomes and pollution
sectors are discussed in detail. In summary, air masses affecting the site from the west, between



north- and south-west, are considered to be representative of the regional background, since they move over a sparsely populated region without any large point sources. In the sector between north and north-east from Welgegund lays the western limb of the Bushveld Igneous Complex, which holds eleven pyrometallurgical smelters (most commonly related to the production of Cr, Fe, V and Ni) within a ~55 km radius , in addition to other industrial, mining and residential sources. In the north-east to eastern sector, the Johannesburg-Pretoria (Jhb-Pta) conurbation is situated, which is inhabited by more than 10 million people, making it one of the forty largest metropolitan areas in the world. In the sector between east and south-east from Welgegund is the Vaal Triangle region, where most of the South African petrochemical and petrochemically-related industries are located, together with other large point sources, such as two coal-fired power stations (without de-SOx and de-NOx) and large pyrometallurgical smelters. Welgegund is also affected by the Mpumalanga Highveld in the eastern sector (indicated by MP in Figure 1). In this region, there are 11 coal-fired power stations (without de-SOx and de-NOx technologies) with a combined installed generation capacity of ca. 46 GW, as well as a very large petrochemical plant, several pyrometallurgical smelters and numerous coal mines, all within a ca. 60 km radius. Furthermore, Welgegund is also affected by air masses passing over the pyrometallurgical smelters in the eastern limb of the Bushveld Igneous Complex situated north-east from Welgegund in the Limpopo Province (indicated by LP in Figure 1).

**Insert Figure 1**

## 2.2 Sampling and analysis

Aerosol samples were collected for one year from 24 November 2010 until 28 December 2011. A Dekati (Dekati Ltd., Finland) PM$_{10}$ cascade impactor (ISO23210) equipped with PTFE filters was used to collect different particulate size ranges, i.e. PM$_{2.5-10}$ (aerodynamic diameter ranging between 2.5 and 10 μm), PM$_{1-2.5}$ (aerodynamic diameter ranging between 1 and 2.5 μm) and PM$_1$ (aerodynamic diameter <1 μm). The pump flow rate was set at 30 L min$^{-1}$. Samples were collected continuously for one week, after which filters were changed. A total of 54 samples were collected for the 54-weeks sampling period for each of the three size ranges. The trace


metals in the PM collected on the 216 PTFE filters were extracted by hot acid leaching (20 ml
HNO$_3$ and 5 ml HCl) and diluted in deionised water (18.2 MΩ) up to 100 mL for subsequent
analysis with an inductively coupled plasma mass spectrometer (ICP-MS). In total, 31 trace
metals could be detected with ICP-MS analysis, which included Na, Mg, Al, K, Ca, Ti, Cr, Mg,
Fe, As, Ba, Cd, Cu, Ni, Zn, V, Mo, Hg, Pb, manganese (Mn), cobalt (Co), platinum (Pt),
beryllium (Be), boron (B), selenium (Se), palladium (Pd), barium (Ba), gold (Au), thallium
(Tl), antimony (Sb) and uranium (U). Trace metal concentrations below the detection limit of
the ICP-MS were considered to have concentrations half the detection limit of the species
considered. This is a precautionary assumption that is frequently used in health-related
environmental studies (e.g. Van Zyl et al., 2014).
**2.3   Statistical analysis**
In an attempt to identify possible sources of trace metals detected, PCFA with Varimax rotation
(v. 13.0 SPSS Inc., Chicago, IL, USA) was performed on the dataset. PCFA has been used
widely in receptor modelling to identify major source sectors. The technique operates on
sample-to-sample fluctuations of the normalised concentrations. It does not directly yield
concentrations of species from various sources, but identifies a minimum number of common
factors for which the variance often accounts for most of the variance of species (e.g. Van Zyl
et al., 2014 and references therein). The trace metal concentrations determined for the 32
species in all three size fractions were subjected to multivariate analysis of Box-Cox
transformation and Varimax rotation, followed by subsequent PCFA. In addition, Spearman
correlations were also performed in order to establish correlations between trace metals in order
to substantiate results obtained with PCFA.
**3   Results**
**3.1   Size-resolved trace metal concentrations**
Although nitric digestion is commonly used to extract and dissolve metals for ICP-MS analysis,
it is unable to dissolve and extract silicate minerals. Therefore Si could not be quantified in this
study. In addition, this limitation of the nitric digestion could also result in determining lower
concentrations of metals associated with the silicate component such as Al and K. It is



estimated that approximately only 7 % Si and 30 % Al is extracted by nitric acid leaching (Ahn
et al., 2011). Therefore, since Si and Al are considered to be the most abundant crustal elements
after oxygen, the trace metal concentrations presented in this paper should be related to the
limitation of nitric digestion, i.e. Si-Al-K components missing from the digestions phase.
Silicate minerals can be dissolved in a mixture of aqua regia and hydrofluoric acid. However,
this is a very difficult procedure, which results in the formation of gaseous $SiF_3$ that is not
determinable by ICP-MS.
In Figure 2, the combined trace metal concentrations in all three size fractions (Figure 2 (a)),
as well as concentrations of the trace metals determined in each of the size fractions are
presented (Figure 2 (b), (c) and (d)). Hg and Ag concentrations were below the detection limit
of the analytical technique for the entire sampling period in all three size fractions and the
concentrations of these species are therefore excluded from Figure 2.
**Insert Figure 2**
The highest median concentration was determined for atmospheric Fe, i.e. 1.4 $\mu g\ m^{-3}$, while
Ca was the second most abundant species with a median concentration of 1.1 $\mu g\ m^{-3}$. Fe
concentrations were significantly higher compared to the other trace metal species determined
at Welgegund. Cr and Na concentrations were the third and fourth most abundant species,
respectively. The median Cr concentration was 0.54 $\mu g\ m^{-3}$, while the median Na level was
0.39 $\mu g\ m^{-3}$. Relatively higher concentrations were also determined for Al, B, Mg, Ni and K
with median concentrations of 0.20 $\mu g\ m^{-3}$, 0.30 $\mu g\ m^{-3}$, 0.18 $\mu g\ m^{-3}$, 0.02 $\mu g\ m^{-3}$ and 0.18 $\mu g$
$m^{-3}$, respectively. The combined atmospheric concentrations of the other trace metals in all the
size fractions were clearly lower.
A comparison of the trace metal concentrations in the three size fractions indicates that Fe and
Ca were the most abundant species in all three size fractions. Fe had the highest median
concentration in the $PM_1$ size fraction, i.e. 0.63 $\mu g\ m^{-3}$, while Ca had the highest median
concentrations in the $PM_{1-2.5}$ and $PM_{2.5-10}$ size fractions, i.e. 0.39 $\mu g\ m^{-3}$ and 0.29 $\mu g\ m^{-3}$,
respectively. The median concentration of Fe in the $PM_1$ was significantly higher compared to
the median concentrations thereof in the $PM_{1-2.5}$ and $PM_{2.5-10}$ size fractions. The third and fourth





most abundant species in all three size fractions were Cr and Na, respectively. Relatively higher
concentrations were also determined for Al, B, Mg, Ni and K in all three size fractions. With
the exception of Fe concentrations in the $PM_1$ size fraction, the concentrations of each of the
trace metal species were similar in all size fractions.
A major source of the trace metal species with elevated levels in all three size fractions can be
considered to be wind-blown dust. Trace metal species typically associated with wind-blown
dust include Fe, Ca, Mg, Al and K. As mentioned, Welgegund is a regional background
location affected by air masses passing over large pollutant source regions and a relatively
clean background area (Figure 1). It is therefore expected that wind-blown dust could have a
major impact on atmospheric trace metal concentrations. In addition, the western Bushveld
Igneous Complex is a major source region affecting Welgegund, with a large number of
pyrometallurgical smelters and mining activities (Tiitta et al., 2014; Jaars et al., 2014). This
source region could contribute to regional elevated levels of Fe, Cr, Ni, Zn, Mn and V measured
at Welgegund. The possible sources of trace metal species measured at Welgegund will be
further explored in section 3.5.

## 3.2   Size distribution of trace metals

In Figure 3, the mean size distributions of each of the trace metal species identified above the
detection limit in the three size fractions are presented. Ti had significantly higher contribution
(80%) in the $PM_{2.5-10}$ size fraction, while Al and Mg also had relatively higher contributions
(~50 and 45%, respectively) in the $PM_{2.5-10}$ size fraction. The $PM_{2.5-10}$ size fraction is usually
associated with wind-blown dust typically comprising Al, Fe, Na, Mg and Ti (Polidori et al.,
2009). 70% or more of all the other trace metal species detected were in the two smaller size
fractions, with approximately 35 to 60% occurring in the $PM_1$ size fraction. The presence of
these trace metal species predominantly in the smaller size fractions, especially considering the
relatively large contribution in the $PM_1$ size fractions, indicates the influence of industrial (high
temperature) activities on air masses measured at Welgegund. However, the large influence of
wind-blown dust on trace metal concentrations determined at Welgegund is also reflected with
approximately 30% of most of these trace metals being present in the $PM_{2.5-10}$ size fraction.
Trace metal concentrations measured at Marikana (van Zyl et al., 2014) indicated that Cr, Mn,



V, Zn and Ni occurred almost exclusively in the PM$_{2.5}$ size fraction, with no contribution by
coarser particles.
**Insert Figure 3**
**3.3    Comparison to previous studies and ambient air quality standards**
In Table 1, the annual average PM$_{10}$ trace metal concentrations determined in this study are
compared to trace metal concentrations determined in other studies. Although the aerosol
sampling periods and frequencies for most of these previous trace metal studies were not
similar to the aerosol sampling period and frequency in this investigation, these results could
be utilised to contextualise the trace metal concentrations. As mentioned previously, Hg and
Ag concentrations were below the detection limit of the analytical technique for the entire
sampling period in all three size fractions. Therefore, concentrations presented for these species
are most likely to be an over estimate due to the precautionary assumption.
**Insert Table 1**
The annual mean PM$_{10}$ trace metal concentrations at Welgegund (Table 1) were typically lower
than previous studies conducted in South Africa (Kgabi, 2006; Kleynhans, 2008; Van Zyl et
al., 2014). This is expected, as Welgegund is a regional background location and the previous
studies were conducted at sites within two priority areas, as mentioned previously. These sites
were also located in two of the major source regions influencing air masses arriving at
Welgegund. Marikana (Van Zyl et al., 2014) and Rustenburg (Kgabi, 2006) are situated
approximately 100 km north-north-west from Welgegund within the western Bushveld Igneous
Complex source region, while the site in the Vaal Triangle (Kleynhans, 2008) source region is
situated approximately 90 km east from Welgegund.
Fe was also the most abundant species at Marikana and Rustenburg, with significantly higher
concentrations compared to Welgegund. Mg was the second most abundant species at
Marikana, with Mg concentrations being an order of magnitude higher than levels thereof at




Welgegund, while Mn and Cr concentrations were the second and third highest, respectively
at Rustenburg. Cr levels at Rustenburg were approximately 2.5 times higher than levels thereof
at Welgegund. However, Cr concentrations measured at Welgegund were approximately two
times higher compared to Cr levels determined at Marikana, which could be attributed to the
contribution of Cr units from wind-blown mineral dust at Welgegund. Ni and Zn concentrations
at Welgegund were an order of magnitude lower compared to levels thereof at Marikana and
Rustenburg. Mn and V concentrations determined at Welgegund were significantly lower
compared levels thereof measured at Rustenburg. V levels measured at Marikana were similar
to concentrations at Marikana, while Mn levels were two times higher at Marikana. Similar to
Welgegund, Na, B and Al were also relatively abundant at Marikana with concentrations of
these species an order of magnitude higher at Marikana. Ca concentrations determined at
Welgegund were similar to the levels thereof determined at Marikana, while K levels were
three times higher at Marikana.
Atmospheric Na had the highest concentrations in the Vaal Triangle, while Fe and K were the
second and third most abundant species, respectively. Fe concentrations were similar at Vaal
Traingle than levels thereof at Welgegund, while the annual average Na concentration was
seven times higher and the annual average K level was an order of magnitude higher at the
Vaal Triangle. In addition, Mg concentrations were approximately five times higher in the Vaal
Triangle. Cr, Ni and Zn that are typically associated with pyrometallurgical industries were
significantly lower in the Vaal Triangle compared to levels thereof at Welgegund. However,
Mn concentrations at the Vaal Triangle were higher compared to levels thereof at Welgegund
and Marikana. This can be attributed to the presence of a ferromanganese (FeMn) smelter in
the Vaal Triangle region, as indicated in Figure 1.
The atmospheric trace metal concentrations determined at Welgegund were also compared to
measurements at regional background sites near Beijing, China (Duan et al., 2012), the west
coast of Portugal (Pio et al., 1996) and Spain (Querol et al. 2007). Al concentrations near
Beijing were significantly higher compared to other trace metal species, while Na was the
second most abundant species. Elevated levels of K, Fe and Ca were also determined near
Beijing. Al, Na and K concentrations were an order of magnitude higher compared to levels of
these species determined at Welgegund. Fe levels were twice as low near Beijing, while Mg
concentrations were three times higher. Ca, Pb and Mn concentrations at Welgegund were
similar to levels thereof near Beijing. All the other trace metal species measured near Beijing





were an order or two orders of magnitude lower compared to concentrations of these species
at Welgegund. Annual average trace metal concentrations determined at the two European
regional background sites were an order or two orders of magnitude lower compared to trace
metal levels determined at Welgegund. The generally lower trace metal concentration
determined at these sites in China and Europe compared to Welgegund can be attributed to the
sites in China and Europe being more removed from a conglomeration of metal sources.
Also indicated in Table 1 are the existing ambient air quality guidelines and standard limit
values for trace metal species prescribed by the WHO air quality guidelines for Europe (WHO,
2005), the European Commission Air Quality Standards (ECAQ, 2008) and the South African
National Air Quality Standards of the South African Department of Environmental Affairs
(DEA) (Government Gazette, 2009). There are currently only guidelines and standards for
seven trace metal species, of which each of the above-mentioned institutions only prescribe
limit values for some of these trace metal species. Comparison of the annual average trace
metal concentrations determined at Welgegund with the annual average standard limit values
indicates that Ni and As exceeded standard limits set by the European Commission of Air
Quality Standards. The annual average Ni concentration of 0.079 µg m$^{-3}$ were approximately
four times higher than the European standard limit value of 0.02 µg m$^{-3}$, while the annual
average As level of 0.0084 µg m$^{-3}$ marginally exceeded the annual standard limit of 0.006 µg
m$^{-3}$. These exceedances can most probably be ascribed to the regional impacts of
pyrometallurgical activities in the Bushveld Igneous Complex. Van Zyl et al. (2014) indicated
that the exceedance of Ni at Marikana situated within the western Bushveld Igneous Complex
could be attributed to base metal refining.
The WHO guideline of 2.5x10$^4$ µg m$^{-3}$ listed for Cr is only for atmospheric concentrations of
Cr(VI) with a lifetime risk of 1:1 000 000. The 0.50 µg m$^{-3}$ annual average Cr concentration
determined can therefore not be compared to the guideline, since this value represents the total
atmospheric Cr concentrations in all the oxidation states. V only has a 24-hour standard limit
value. Therefore, V concentrations determined in this study cannot directly be compared to this
standard limit. However, the 24-hour average calculated from the highest weekly V
concentration (0.084 µg m$^{-3}$) was 0.012 µg m$^{-3}$, which was two orders of magnitude lower than
the 24-hour V standard limit of the European Commission Air Quality Standards.





Since Pb is the only trace metal for which a South African ambient air quality standard limit
exists, it must also be noted that Pb concentrations did not exceed any standard limit. The
annual average Pb concentrations determined at Welgegund (0.0078 µg m$^{-3}$) were an order of
magnitude lower than levels thereof at Marikana and Vaal Triangle, and three orders of
magnitude lower than Pb levels determined at Rustenburg. However, the annual average Pb
concentrations at Vaal Triangle, Marikana and Rustenburg were below the standard limit
(Kleynhans, 2008; Van Zyl et al., 2014; Kgabi, 2006). These low Pb concentrations can be
partially ascribed to de-leading of petrol in South Africa. Furthermore, Pb concentrations
determined at Beijing were similar to levels thereof determined at Welgegund.
Since the measurement of the ambient Hg concentrations is receiving increasing attention in
South Africa and it is foreseen that a standard limit value for Hg levels will be prescribed in
the near future, it is also important to refer to the Hg concentrations that were below the
detection limit of the analytical instrument for the entire sampling period. Van Zyl et al. (2014)
also indicated that Hg was below the detection limit of the analytical technique for aerosol
samples collected at Marikana. This can be expected, since particulate Hg only forms a small
fraction of the total atmospheric Hg, with Hg being predominantly present in the atmosphere
as gaseous elemental Hg (GEM) (Venter et al., 2015, Slemr et al., 2011).

### 3.4 Seasonal trends

The climate and weather of South Africa is characterised by its distinctive wet and dry seasons,
which have an influence on concentrations of atmospheric species (Tyson and Preston-Whyte,
2000). Therefore, in Figure 4, the total concentrations of the trace metal species in the PM$_1$ (a),
PM$_{1-2.5}$ (b) and PM$_{2.5-10}$ (c) size fractions measured at Welgegund for each month are presented,
with the contributing concentrations of each of the trace metals indicated. In the PM$_{1-2.5}$ and
PM$_{2.5-10}$ size fractions relatively higher total trace metal concentrations are observed from
August to December. These periods coincided with the end of the dry season, which occurs in
this part of South Africa typically from mid-May to mid-October (e.g. Tyson and Preston-
Whyte, 2000). The end of the dry season is typically characterised by increases in wind speed
in August (e.g. Tyson and Preston-Whyte, 2000). Therefore, these elevated trace metal
concentrations determined in the PM$_{1-2.5}$ and PM$_{2.5-10}$ size fractions can partially be attributed
to decreased wet removal in conjunction with increases in wind generation thereof. The PM$_1$





size fractions also had relatively higher during the end of dry season period, especially during
September and October. However, slightly higher trace metal concentrations are also observed
in the $PM_1$ size fraction in the austral winter months from June to August. This can be ascribed
to the presence of more pronounced inversion layers during this time of the year (e.g. Tyson
and Preston-Whyte, 2000) that trap pollutants near the surface, which signifies the contribution
of industrial sources to $PM_1$ species.

8        **Insert Figure 4**

The monthly concentrations of each of the trace metal species determined in the $PM_1$ and $PM_{1-}$
$_{2.5}$ size fractions reveal the highest contributions from Fe and Ca in both these size fractions for
each of the months. The concentrations of Na and Cr that were the third and fourth most
abundant species, respectively, as well as the elevated levels of Al, B, Mg, Ni and K are also
reflected in the monthly distributions in the $PM_1$ and $PM_{1-2.5}$ size fractions. However, although
Fe and Ca were slightly higher in the $PM_{2.5-10}$ size fraction, a more even contribution from the
concentrations of Fe, Ca, Na, Cr, Al, B, Mg, Ni and K is observed (with the exception of
November as mentioned previously). This can be attributed to species in this larger size fraction
consisting predominantly of wind-blown dust (Adgate et al., 2007) with no additional industrial
sources of these species.
**3.5   Source apportionment**
As a first approach in the source apportionment investigation, Spearman correlation diagrams
were prepared for each size fraction. In Figure 5, Spearman correlations of the $PM_1$, $PM_{1-2.5}$
and $PM_{2.5-10}$ size fractions are presented, i.e. Figures 5a, 5b and 5c, respectively. From Figure
5 relatively good correlations is observed between trace metals associated with
pyrometallurgical activities, i.e. Fe, Cr, Zn, Mn and V in all three size fractions. Na, Mg and
Ca also correlate with each other in all three size fractions, indicating the crustal (earth)
influence. Relatively good correlations are also observed between Ti and crustal species in the
$PM_{2.5-10}$ size fraction. In addition, these crustal species (Na, Mg, and Ca) also correlate with
species associated with pyrometallurgical activities (Fe, Cr, Zn, Mn and V). As mentioned in





Sections 3.1 and 3.2, although the influence of the pyrometallurgical smelters in the western
Bushveld Complex is evident, the large influence of wind-blown dust on trace metal
concentrations determined at Welgegund is also reflected with approximately 30% of most of
the trace metals being present in the $PM_{2.5-10}$ size fraction.
**Insert Figure 5**
In an effort to determine sources of trace metals, PCFA was applied as an exploratory tool,
since much larger datasets are required for definitive source apportionment with PCFA.
Therefore, only the most apparent groupings of metal species relating to expected sources in
the region were identified. PCFA of the $PM_{1-2.5}$ and $PM_{2.5-10}$ size fractions did not reveal any
meaningful factors. This was attributed to the large influence of wind-blown dust on trace
metals measured at Welgegund with all the factors obtained for the $PM_{1-2.5}$ and $PM_{2.5-10}$ size
fractions containing mostly crustal species loadings. In Figure 6, the factor loadings obtained
for the $PM_1$ size fraction are presented indicating four statistically significant factors with
eigenvalues equal to or greater than one (Pollisar et al., 1998). These four factors obtained
explained 88% of the variance.
**Insert Figure 6**
Factor 1 explained 59.6 % of the total system variance and was mainly loaded with trace metal
species that are typically associated with wind-blown dust, i.e. Ca, Fe, Na, Mg and Al (Adgate
et al., 2007). Therefore, this factor was identified as the crustal factor. The contribution of small
metal ore units from wind-blown dust is also reflected in this factor with a relatively high
loadings of species such as V, Mn, Zn and Cr. Mn is present in most of the ores from which
metals are produced in the western Bushveld Igneous Complex. The smaller contribution from
Mn compared to Fe in this factor is also indicative of wind-blown dust, since Mn is more
volatile than Fe (Kemink, 2000). Therefore, a higher contribution is expected from Mn
compared to Fe from pyrometallurgical sources.



Factor 2 and 3 explained 16.5 and 4.3 % of the variance in the data, which was identified as
pyrometallurgical-related factors. Factor 2 revealed higher loadings of Cr, Fe Mn, Ni and Cu,
while Factor 3 was predominantly loaded with Cr, Fe and V. Fe and Cr are associated with the
large number of ferrochromium smelters in the Bushveld Igneous Complex, while Ni related
to base metal smelters that refine base metals extracted from the PGM production processes.
In addition, Al present in Factor 2 is may be associated with fly ash formed during high
temperature processes, which include coal combustion. It must be noted that coal fly ash has a
composition, which is rather similar to that of crustal material (Mouli, et al., 2006). Mn has a
substantially lower vapour pressure than most of the heavy metals produced in this region.
Therefore, the coincidental influence of the pyrometallurgical industries is reflected by the high
loadings of Mn and Ni in Factor 2.
Factor 4 was considered to be indicative of trace metal species associated with slimes dams
from Au mining and recovery in the region, which is especially signified by the U and Au
loadings in this factor. In addition, this factor is mostly loaded with the metal species for which
significantly lower concentrations were measured. This factor explained 7.6 % of the total
system variance.
Pollution roses of each of the trace metal species detected were also compiled in an effort to
substantiate the sources identified with PCFA for the $PM_1$ size fraction, as well as to verify the
influence of wind-blown dust that contributed to obtaining no meaningful factors for $PM_{1-2.5}$
and $PM_{10-2.5}$. In Figure 7, these pollution roses are presented, which indicate higher trace metal
concentrations associated with wind directions from the north to western sector from
Welgegund for all the trace metal species. As mentioned previously, the north to south-western
sector from Welgegund is considered to be a relatively clean region without any large pollutant
sources. Therefore, the most significant source of atmospheric trace metal species originating
from this sector can be considered to be wind-blown dust (e.g. from the Karoo and Kalahari).
This is also indicated by the higher atmospheric concentrations of specifically Ca, Fe, Na, Mg,
Al and Ti associated with the north-western sector. Furthermore, the concentrations of trace
metal species originating from the north can also be associated with pyrometallurgical
industries in the western Bushveld Igneous Complex. The influence of these activities is
reflected by the relatively higher concentrations of Cr, Ni, Mn, V and As associated with winds
originating in the north. It is also evident form these pollution roses that atmospheric Fe




concentrations have contributions from wind-blown dust from the north-western sector, as well
as from pyrometallurgical activities in the north.
**Insert Figure 7**
**4    Conclusions**
Of the elements analysed in the aerosol samples, atmospheric Fe had the highest concentrations
in all three size fractions, while Ca was the second most abundant species.  Cr and Na
concentrations were the third and fourth most abundant species, respectively, while relatively
higher concentrations were also determined for Al, B, Mg, Ni and K. With the exception of Fe
that had higher concentrations in the $PM_1$ size fraction, the concentrations of the trace metal
species in all three size ranges were similar. With the exception of Ti, Al and Mg, 70% or more
of the trace metal species detected were in the two smaller size fractions, which indicated the
influence of industrial activities on trace metals measured at Welgegund. However, the large
influence of wind-blown dust on trace metal concentrations determined at Welgegund is
reflected by 30% and more of trace metals being present in the $PM_{2.5-10}$ size fraction
A comparison of trace metal concentrations determined at Welgegund with trace metal
measurements conducted in the western Bushveld Igneous Complex (Kgabi, 2006; van Zyl et
al., 2014) indicated that Fe was also the most abundant species, while other trace metals
determined at Welgegund were also measured in the western Bushveld Igneous Complex.
However, concentrations of these trace metal species were significantly higher in the western
Bushveld Igneous Complex. Trace metal concentrations were also compared to levels thereof
in the Vaal Triangle (Kleynhans, 2008) where. Fe concentrations were similar to levels thereof
at Welgegund, while concentrations of species associated with pyrometallurgical smelting
were lower. Comparison to atmospheric trace metal species measured at international
background sites indicated that trace metal concentrations at Welgegund were generally lower,
with the exception of Al, Na and K concentrations measured at Beijing, China (Duan et al.,
2012) that were an order of magnitude higher. Annual average Ni (0.079 $\mu g\ m^{-3}$) were four
times higher than the European Commission Air Quality Standards limit value, which could
possibly be attributed to the influence of base metal refining in the western Bushveld Igneous



Complex. As marginally exceeded the European Commission Air Quality Standards limit
value, which also reflects the regional impacts of pyrometallurgical industries.
Al three size fractions indicated elevated trace metal concentrations coinciding with the end of
the dry season. This could partially be attributed to decreased wet removal and increases in
wind generation of particulates.
PCFA analysis revealed four statistically significant factors in the $PM_1$ size fraction, i.e. crustal,
pyrometallurgical-related and Au slimes dams. No meaningful factors were determined for the
$PM_{1-2.5}$ and $PM_{2.5-10}$ size fractions, which were attributed to the large influence of wind-blown
dust on atmospheric trace metals determined at Welgegund. Pollution roses confirmed this
influence of wind-blown dust on trace metal concentrations, while the impact of industrial
activities was also substantiated.
**5  Acknowledgements**
The financial assistance of the National Research Foundation (NRF) towards this research is
hereby acknowledged. Opinions expressed and conclusions arrived at are those of the author
and are not necessarily to be attributed to the NRF. V. Vakkari wishes to acknowledge financial
support by the Academy of Finland Center of Excellence program (grant no. 272041).

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





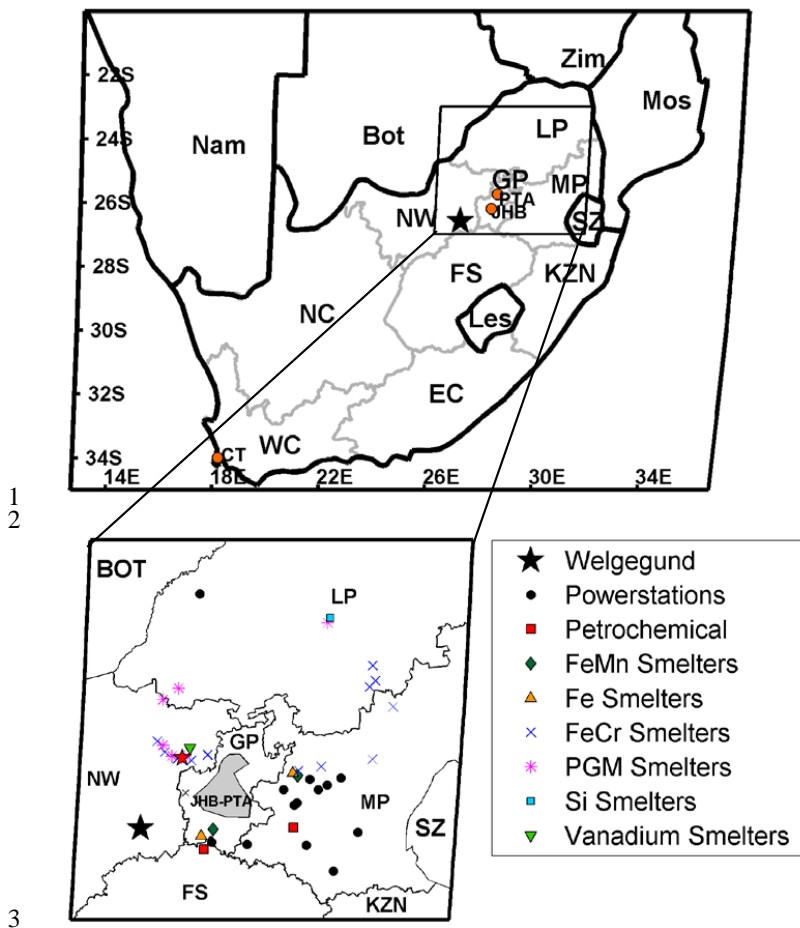

Figure 1: Geographical map indicating Welgegund (black star), as well as the major point
sources and the Johannesburg-Pretoria (JHB-PTA) conurbation. Neighbouring countries to
South Africa (Nam = Namibia, Bot = Botswana, Zim = Zimbabwe, Mos = Mozambique, SZ
= Swaziland, Les = Lesotho) as well as South African provinces (LP = Limpopo , NW =
North-West, FS = Free State, KZN = Kwa-Zulu Natal, MP = Mpumalanga, NC = Northern
Cape, EC = Eastern Cape and WC = Western Cape) are also indicated.





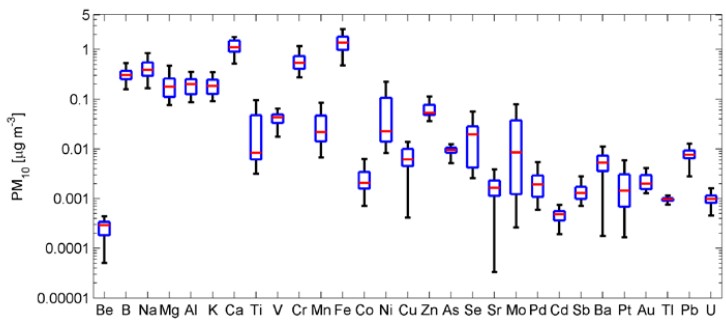

1                                                      (a)

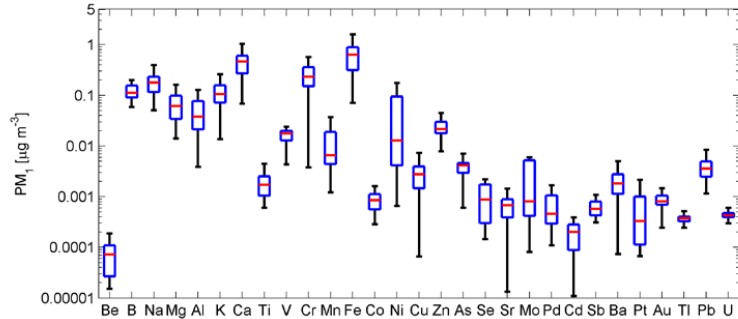

2                                                      (b)

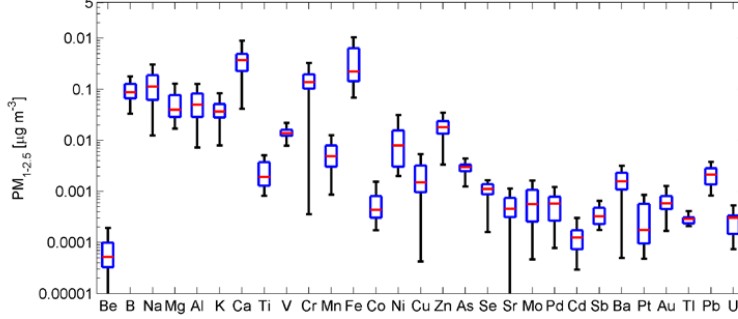

3                                                      (c)





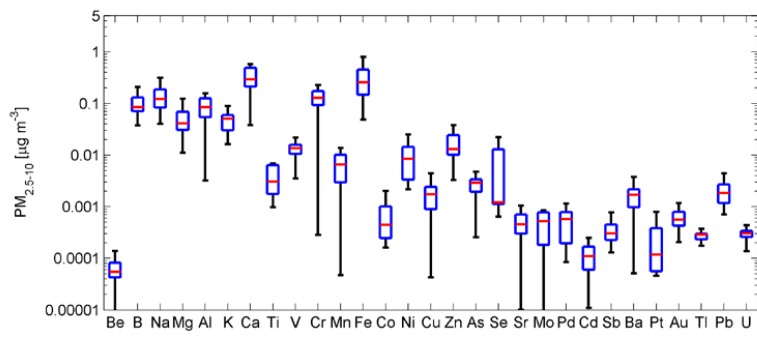

1                                                                                                      (d)

2       Figure 2: Box and whisker plots of trace metal concentrations in the (a) $PM_{10}$ (sum of trace

3       metal concentrations in the three size fractions), (b) $PM_1$, (c) $PM_{1-2.5}$, and (d) $PM_{2.5-10}$ size

4       fractions. The red line indicates the median concentrations, the blue rectangle of the boxplot

5       represents the $25^{th}$ and $75^{th}$ percentiles, while the whiskers indicate $\pm 2.7$ times the standard

6       deviation





Table 1: Annual mean PM$_{10}$ trace metal concentrations measured at Welgegund, annual
average standard limits, as well as annual average trace metal levels determined in other
studies in South Africa, China and Europe. Concentration values are presented in µg m$^{-3}$

| PM$_{10}$ annual average | ICP detection limits (x10$^{-5}$) | Welgegund (This study) | Annual standard limit | South Africa | | | Beijing, China (Duan et al., 2012) | West coast of Portugal (Pio et al., 1996) | Spain (Querol et al., 2007) |
| --- | --- | --- | --- | --- | --- | --- | --- | --- | --- |
| | | | | Marikana (Van Zyl et al., 2014) | Rustenburg (Kgabi, 2006) | Vaal Triangle (Kleynhans, 2008) | | | |
| Be | 0.293 | **0.0002** | | 0.020 | | | 0.100 | | <0.001 |
| B | 4.415 | **0.28** | | 1.300 | | | | | |
| Na | 8.515 | **0.38** | | 1.410 | | 2.800 | 1.450 | | |
| Mg | 3.504 | **0.23** | | 2.040 | | 1.000 | 0.637 | | |
| Al | 6.960 | **0.17** | | 1.280 | | | 2.180 | 0.200 | |
| K | 12.98 | **0.14** | | 0.680 | | 1.300 | 1.170 | | |
| Ca | 19.88 | **1.1** | | 1.080 | | | 0.996 | | |
| Ti | 5.729 | **0.072** | | 0.120 | 0.180 | 0.020 | 0.069 | | 0.019 |
| V | 1.736 | **0.037** | 1.000$^{(b)\#}$ | 0.040 | 0.160 | | | <0.001 | 0.005 |
| Cr | 0.233 | **0.50** | 2.5x10$^{4(a)*}$ | 0.240 | 1.370 | 0.050 | 0.022 | <0.001 | 0.001 |
| Mn | 2.064 | **0.026** | 0.15$^{(a)}$ | 0.060 | 4.390 | 0.120 | 0.036 | 0.002 | 0.005 |
| Fe | 15.86 | **1.2** | | 2.540 | 9.760 | 1.280 | 1.090 | 0.028 | |
| Co | 0.8146 | **0.0035** | | 0.140 | | | <0.001 | | <0.001 |
| Ni | 4.000 | **0.079** | 0.020$^{(b)}$ | 0.330 | 0.770 | 0.040 | 0.020 | <0.001 | 0.003 |
| Cu | 3.529 | **0.0069** | | 0.180 | 0.210 | 0.050 | 0.010 | 0.003 | 0.008 |
| Zn | 14.13 | **0.053** | | 0.490 | 0.340 | 0.090 | 0.027 | 0.003 | 0.026 |
| As | 4.730 | **0.0084** | 0.006$^{(b)}$ | 0.260 | | | 0.003 | 0.002 | <0.001 |
| Se | 10.51 | **0.0074** | | 0.580 | | | 0.001 | <0.001 | 0.001< |

Table 1: continued...





| | | | | | | | | | |
|---|---|---|---|---|---|---|---|---|---|
| **Sr** | 0.819 | **0.0017** | | | | | 0.010 | | 0.005 |
| **Mo** | 0.421 | **0.015** | | | | | 0.007 | | 0.004 |
| **Pd** | 7.394 | **0.0018** | | 0.410 | | | | | |
| **Ag** | 1.030 | ***0.0005*** | | | | | <0.001 | | |
| **Cd** | 0.637 | **0.0004** | 0.005 (a)(b) | 0.030 | | | <0.001 | <0.001 | <0.001 |
| **Sb** | 0.444 | **0.0013** | | | | | <0.001 | | <0.001 |
| **Ba** | 3.194 | **0.0040** | | 0.140 | | | 0.018 | | <0.008 |
| **Pt** | 6.962 | **0.0016** | | 0.350 | | | | | |
| **Au** | 7.340 | **0.0031** | | 0.380 | | | | | |
| **Hg** | 9.971 | ***0.0002*** | 1.000(a) | 0.550 | | | | | |
| **Tl** | 4.917 | **0.0007** | | 0.270 | | | | | <0.001 |
| **Pb** | 2.592 | **0.0078** | 0.5 (a)(b)(c) | 0.080 | 0.420 | 0.040 | 0.053 | 0.003 | 0.009 |
| **U** | 8.527 | **0.0009** | | | | | | | |

*\* WHO guideline for Cr(VI) concentrations associated with an excess lifetime risk of 1:1 000 000*
*# 24-h limit value*
a) *WHO air quality guidelines for Europe,* b) *European Commission Air Quality Standards, c) National Air*
*Quality Act of the South African Department of Environmental Affairs*



(a)
(b)
(c)
Figure 4: The monthly median trace metal concentrations in the $PM_1$ (a), $PM_{1–2.5}$ (b) and
$PM_{2.5–10}$ (c) size fractions





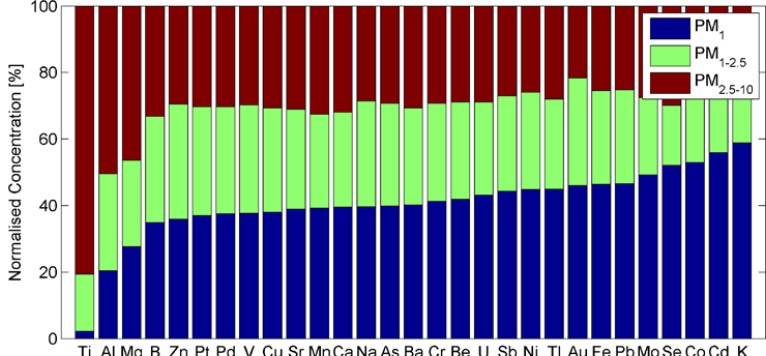

3    Figure 3: Mean size distributions of individual trace metal species detected. Species are
4    arranged by increasing concentration in the PM$_1$ size fraction





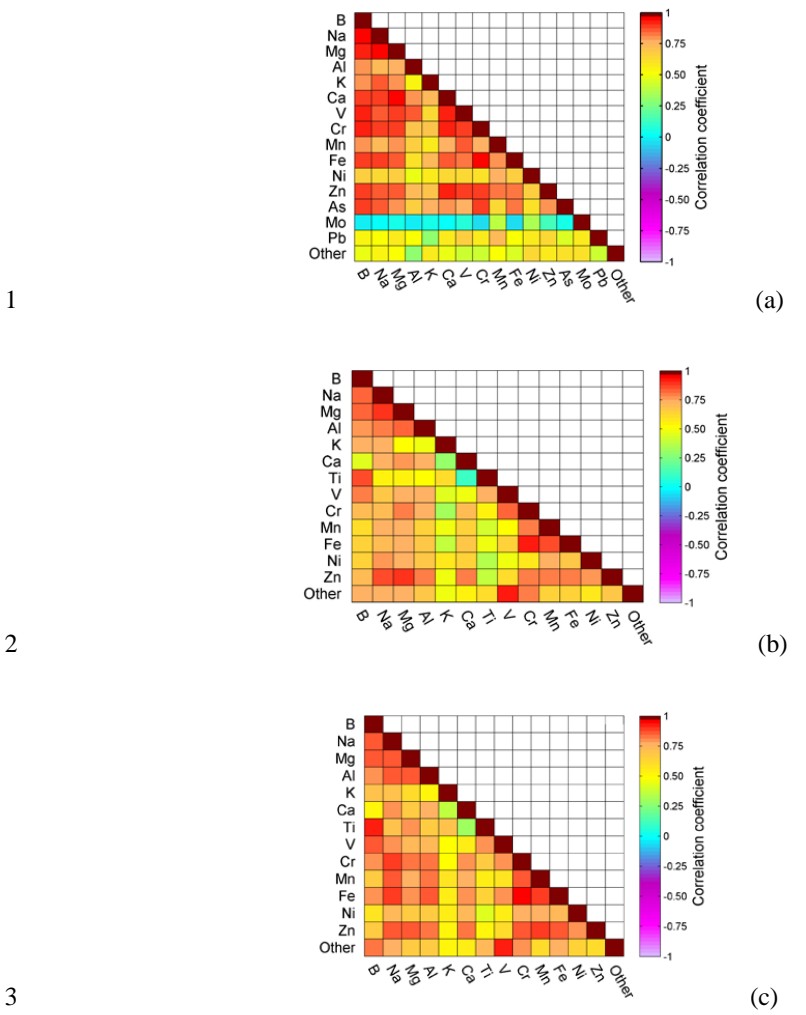

1                                                                                    (a)

2                                                                                    (b)

3                                                                                    (c)

4       Figure 5: Spearman correlations of trace metal species in the $PM_1$ (a), $PM_{1-2.5}$ (b) and $PM_{2.5-10}$

5       (c) size fractions





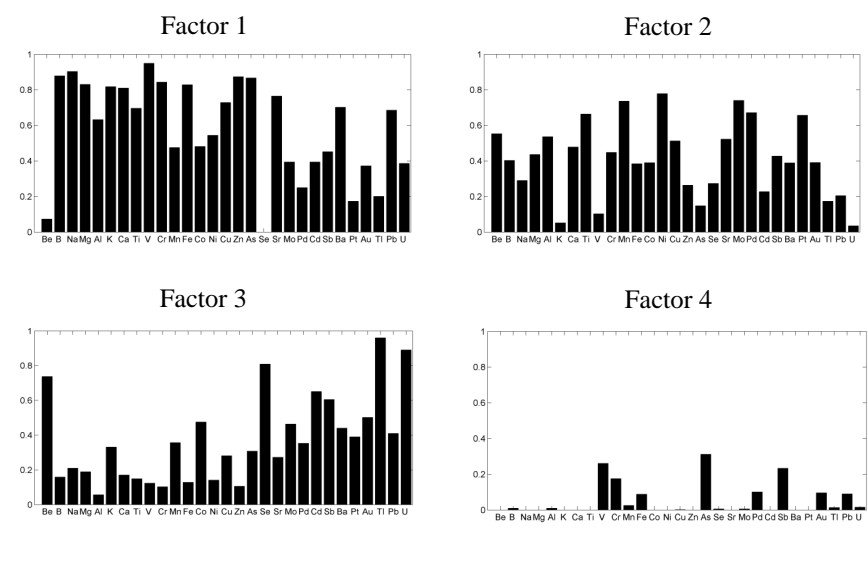

3      Figure 6: PCA/FA of the trace metal concentration in the PM$_1$ size fraction. Four dominant factors are identified.





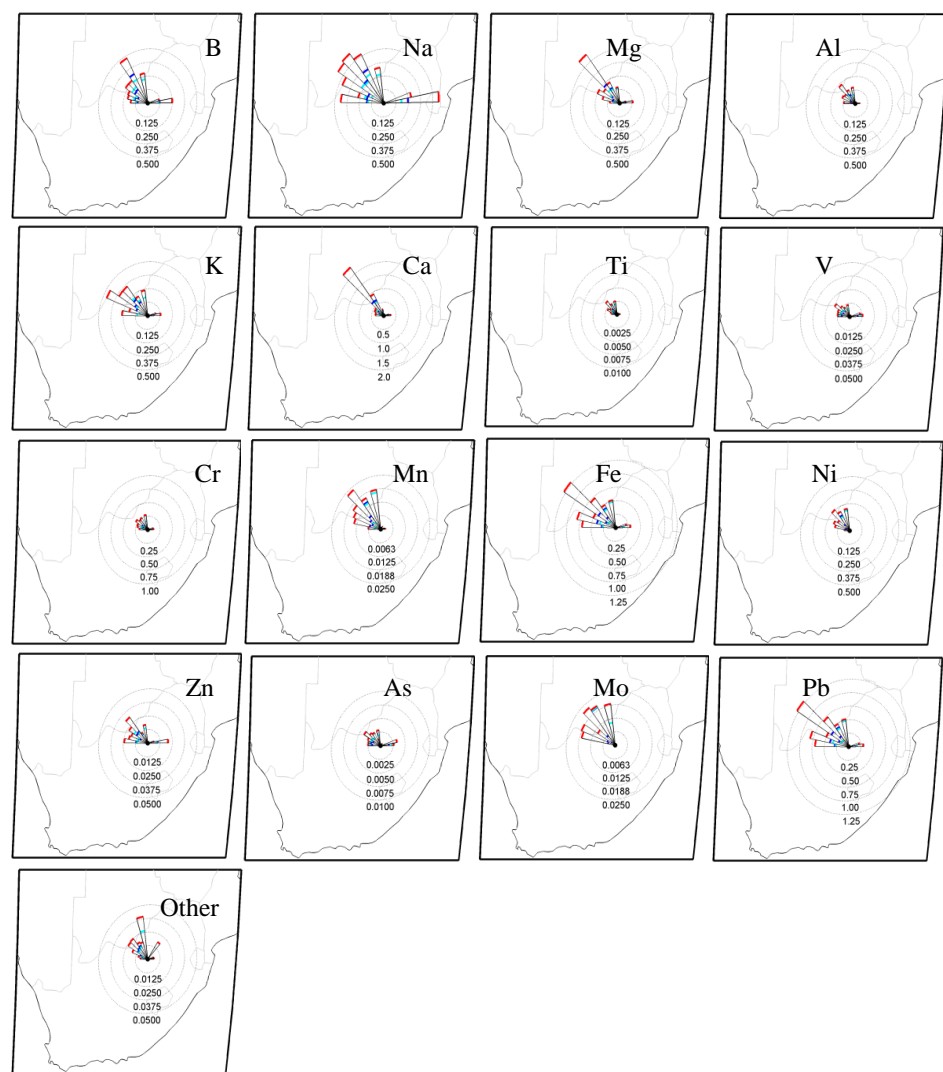

3    Figure 7: Pollution roses of trace metal species that were 25% or more of the time detected with

4    the analytical technique