# Peer review of "Atmospheric trace metals measured at a regional"

_Atmospheric Chemistry and Physics, 2016_

## Referee Comment (RC1) · Anonymous Referee #2 · 11 Nov 2016

General comment:

This study reports trace metal composition of atmospheric particulate matter (aerosols) in three different size fractions (PM1, PM1-2.5 and PM2.5-10) at a regional background site (Welgegund) in South Africa. The reported data present a weekly averaged trace metal composition spanned over a year time. Authors have discussed the variation of different trace metals in various size fractions, their seasonal variability, compared data with several studies and tried to identify sources of different trace metals using statistical tool (PSCF). Overall, the study is OK in a regional/local context presuming paucity of aeolian trace metal composition data from the South African region. However, it lacks global significance and the manuscript appears to be just reporting observations

at the sampling location. Further, I feel, the scientific content is below the requirement of ACP. Thus, I think, this manuscript is not suitable for publication in ACP. Below, I have pointed out few specific comments which may help authors to revise and submit in a different Journals.

Specific comments: 1) Abstract and introduction: Why collection of samples were undertaken at the mentioned site? Why it is called background site? How background site is defined and why it is important to study background site composition?

2) Sampling and Analyses: A mixture of HCL and HNO3 have been used to dissolve (or leach) the trace metals (TM) in this study. So, the metals associated with aluminosilicate phase are underestimated. Authors have mentioned it in start of section 3.1. However, they should mention, several metals e.g. Al, Mg, Ca, Fe, e.t.c are underestimated concentrations especially those samples having high aeolian dust content.

3) Section 3.1 and 3.2 can be merged to a single section and the variability of trace metal composition in various sizes and total TM concentrations can be discussed.

4) Page 8, Line 7-9: How dust is impacting TM concentration? Its not clear. What is the source of dust? It is discussed by the authors that the sampling site is surrounded by pollutant emitting sources at least in the eastern region. However, there is no mention of dust source in the west or even eastern part of sampling site. Is there any hotspot for dust emission in the proximity of sampling site. Or is it local dust?

5) Section 3.3 on comparing data set with previous studies from similar and other area is over discussed. Why air-quality aspect suddenly brought in the discussion. Does this study have any bearing on health issues?

6) Page 10, Line 4-5: Why and how dust can contribute Cr to the particulate matter?

7) Section 3.4: Seasonal trend cannot be discussed based on 1 yr data, however seasonal variability can be.

---

## Referee Comment (RC2) · Anonymous Referee #3 · 10 Jan 2017

General Comments This is an easy to read and well written discussion paper on the analysis of trace metals in aerosol samples collected from a site in central South Africa. The extent and originality of the contribution to the understanding aerosol trace metal contributions are not new globally, but rather across the region, as is highlighted by the authors. The analytical methods employed are well established in literature. The figures/table are clear and evaluation of the generated results and their integration with the existing body of knowledge is sufficient. There is correct use of references and their presentation in a reference list.

Scientific Questions ..Pg 8 Line 24 to 27. This statement should be explored further in this study and supported by clear justification based on the understanding of long range

transportation of pollution in the region. Is this statement supported by e.g., trajectory analysis of air masses to Welgegund? ..Pg 11 Line 14. Throughout the manuscript, the phrase 'standard limit' has been used. Please note that a STANDARD is the limit value. Therefore to avoid redundancy, use STANDARD without limit. ..Pg 11 Section 3.5. For the average concentrations, where are the standard deviations and how significant are they? How would these deviations affect the PCFA receptor modelling results, if at all they do?

..Technical Corrections ..Pg 2, Line 27 should be referenced IPCC, 2014. Consistency in the use of AND or & in references siting ..Pg 3, Line 28 should be referenced WHO, 2005 ..Pg 4 Lines 4, 5 and 6 and Pg 11 Line 11 "Government Gazette" should be defined based on author DEAT or DEA depending on the years. ..Pg 5 Line 11, Use full text on first mention e.g., Desulfurization (DeSOx) / Denitrification (DeNOx) equipment. ..Pg 5 Line 26, Expand ICP-MS on first mention. ..Pg 11 Line 23 Should this be 2.5x10E4 or 10E-4? Please check Table 1 as well. ..For references, check for consistency in the use of DOI/doi throughout the discussion paper.

---

## Author Comment (AC1) · 13 Feb 2017

General comment:

This study reports trace metal composition of atmospheric particulate matter (aerosols) in three different size fractions (PM1, PM1-2.5 and PM2.5-10) at a regional background site (Welgegund) in South Africa. The reported data present a weekly averaged trace metal composition spanned over a year time. Authors have discussed the variation of different trace metals in various size fractions, their seasonal variability, compared data with several studies and tried to identify sources of different trace metals using statistical tool (PSCF). Overall, the study is OK in a regional/local context presuming paucity of aeolian trace metal composition data from the South African region. However, it lacks global significance and the manuscript appears to be just reporting observations at the sampling location. Further, I feel, the scientific content is below the requirement of ACP. Thus, I think, this manuscript is not suitable for publication in ACP. Below, I have pointed out few specific comments which may help authors to revise and submit in a different Journals.

The authors would like to thank Referee #2 for reviewing this manuscript. Although it is recommended by Referee #2 that this paper is not suitable for publication in ACP, the relevance of this study on a regional scale is acknowledged by Referee #2. In another review of this paper, Referee #3 also acknowledged the relevance of this work on a regional scale. Therefore Referee #3 indicated that the extent and originality of this contribution lies within understanding aerosol trace metal contributions on a regional scale, which is highlighted by the authors. South Africa is an understudied region with only a few studies conducted on atmospheric trace metal concentrations that are published in peer-reviewed journals or available in the public domain. Therefore, the relevance of this study, although more on a on a regional scale, is indicated by both referees. However, in an effort to indicate the global relevance of this paper the following sentence was added in the "Introduction" referring to the global significance of the western Bushveld Igneous Complex, which is one of the source regions influencing Welgegund:

"…measurement station in South Africa. Welgegund is a comprehensively equipped regional background atmospheric measurement station that is ~100 km downwind of the most important source regions in the interior of South Africa (e.g. Tiitta et al., 2014). These source regions include the western Bushveld Igneous Complex (situated within the Waterberg-Bojanala Priority Area) where a large number of pyrometallurgical smelters are situated, which can be considered of global importance, e.g.

as a supplier of platinum group metals (PGMs) utilised in automotive catalytic converters and as the dominant global chromium supplying region. In an effort to determine major sources of trace metals…"

Furthermore, Referee #3 pointed out that the analytical methods employed in this study are well established in literature and also indicated that the results presented were adequately integrated with the existing body of knowledge. Therefore in view of the positive review of this paper by Referee #3 recommending possible publication in ACP, the authors believe that the scientific content is on the required level for publication in ACP.

In addition, each of the specific comments made by Referee #2, as well as the scientific questions raised and technical corrections suggested by Referee #3, was also addressed in an effort to further improve the scientific content of this manuscript. All changes in the manuscript are indicated with track changes.

Specific comments:

1) Abstract and introduction: Why collection of samples were undertaken at the mentioned site? Why it is called background site? How background site is defined and why it is important to study background site composition?

In the second last paragraph of the "Introduction" it is indicated that only a few studies on atmospheric trace metals have been conducted for South Africa, of which most of these studies were conducted within highly polluted regions. It is also mentioned that air quality outside these polluted areas could also be adversely affected through regional transport. Therefore in the last paragraph of the "Introduction" it is specified that the aim of this study was to determine atmospheric trace metal concentrations on a regional scale, i.e. at Welgegund, which is a regional background station impacted by the major source regions in the interior of South Africa. One of these source regions is the western Bushveld Igneous Complex where a large number of pyrometallurigical industries are located. Therefore the relevance of measuring samples at a background site is argued within these last two paragraphs in the "Introduction" as follows (versions of these paragraphs as in the revised manuscript):

"South Africa has the largest industrialised economy in Africa, with significant mining and metallurgical activities. South Africa is a well-known source region of atmospheric pollutants, which is signified by three regions being classified through legislation as air pollution priority areas, i.e. Vaal Triangle Airshed Priority Area (DEAT, 2006), Highveld Priority Area (DEAT, 2007) and Waterberg-Bojanala Priority Area (DEA, 2012). Air quality outside these priority areas is often adversely affected due to regional transport and the general climatic conditions, such as low precipitation and poor atmospheric mixing in winter. Only a few studies on the concentrations of atmospheric trace metals in South Africa have been conducted (Van Zyl et al., 2014; Kgabi, 2006; Kleynhans, 2008). In addition,

most of these studies were also conducted within these priority areas containing a significant number of large point sources, and regional impacts of atmospheric trace metals could therefore not be assessed.

In this study, trace metals were determined in three size ranges in aerosol samples collected for one year at the Welgegund atmospheric measurement station in South Africa. Welgegund is a comprehensively equipped regional background atmospheric measurement station that is ~100 km downwind of the most important source regions in the interior of South Africa (e.g. Tiitta et al., 2014). These source regions include the western Bushveld Igneous Complex (situated within the Waterberg-Bojanala Priority Area) where a large number of pyrometallurgical smelters are situated, which can be considered of global importance, e.g. as a supplier of platinum group metals (PGMs) utilised in automotive catalytic converters and as the dominant global chromium supplying region. In an effort to determine major sources of trace metals on a regional scale, source apportionment was also performed by applying principal component factor analysis (PCFA)."

Furthermore, in the first paragraph of Section 2.1 "Site description" a sentence was added to indicate that Welgegund is a background site since there are no large point sources within close proximity of the site. Reference is also made in this paragraph to the map in Figure 1 and the 96-hour overlay trajectories presented as a supplement in Figure S1, which were also compiled in order to address Comment 4, as well as a comment made by Referee #3. This text in this paragraph was changed as follows:

"Aerosol sampling was performed at Welgegund (www.welgegund.org, 26°34'11.23"S, 26°56'21.44"E, 1480 m a.s.l. (above sea level)) in South Africa, which is a regional background station with no large point sources in close proximity. As indicated in Figure 1 and the 96-hour overlay back trajectories presented in Figure S1, Welgegund is situated in the interior of South Africa and is frequently affected by air masses moving over the most important anthropogenic/industrial source regions in the interior (Beukes et al., 2013, Tiitta, et al., 2014, Jaars, et al., 2014, Vakkari et al., 2015; Booyens et al., 2015). Also indicated in Figure 1 are the major industrial point sources,…"

The third sentence in the Abstract was also changed in order to indicate that Welgegund is a background site as follows:

"…the aim of this study was to determine trace metals concentrations in aerosols collected at a regional background site, i.e. Welgegund, South Africa. $PM_1$,…"

2) Sampling and Analyses: A mixture of HCL and HNO3 have been used to dissolve (or leach) the trace metals (TM) in this study. So, the metals associated with aluminosilicate phase are underestimated. Authors have mentioned it in start of section 3.1. However, they should mention, several metals e.g. Al, Mg, Ca, Fe, e.t.c are underestimated concentrations especially those samples having high aeolian dust content.

Although there are deficiencies associated with the analytical method employed, i.e. unable to efficiently dissolve and extract aluminosilicate minerals, these analytical methods are commonly utilised for the analysis of atmospheric trace metals. This paucity was recognised and discussed by the authors in the start of Section 1. Referee #3 also indicated that analytical methods employed were adequate and "well established in literature". Therefore results were interpreted throughout the paper within the limitation of the analytical technique. The text in in the first paragraph of Section 3.1 was changed in order to include additional metals that could be potentially underestimated as indicated by Referee #2 as follows:

"…study. In addition, this limitation of the nitric digestion could also result in determining lower concentrations of metals associated with the silicate component such as Al, K, Mg, Ca and Fe, especially, for samples that have high aeolian dust content. It is estimated that approximately…"

Furthermore, an additional paragraph was added at the end of the "Conclusions" section considering another analytical technique typically applied for analyses of atmospheric trace metals, i.e. X-ray fluorescence (XRF). A paper is referenced where ICP-MS and XRF analysis of atmospheric trace metal were compared, which indicated the benefits and limitations of each method. Therefore a future recommendations is made to conduct both analytical techniques if possible, which should supplement one another:

"There are limitations associated with nitric digestion for ICP-MS analysis employed in this study, which could lead to the underestimation of aluminosilicates and metal species associated with it. X-ray fluorescence (XRF), for instance, is an alternative analytical method that can be used to assess the chemical composition of PM collected on filters. The use of this technique has many advantages, e.g. non-destructive technique, little sample preparation required, relatively low cost per sample. In order to compare XRF with ICP-MS (digestion using ultrasonication in an HF-$HNO_3$ acid mixture) aerosol filter based analyses, Niu et al. (2010) analysed co-located duplicate samples collected in indoor and outdoor environments. Very good correlations for elements present at concentrations above the detection limits of both the ICP-MS and energy dispersive-XRF methods were found. However, much more elements analysed by the ICP-MS technique passed the quality criteria proposed by the afore-mentioned authors, including elements typical for alumina silicates and other wind blow dust compounds that were likely under estimated in the results presented in this paper. Therefore, although the digestion method used in this study is well established, it is recommended that future work should perform digestion using ultrasonication in an HF-$HNO_3$ acid mixture and, if possible, conduct both XRF and ICP-MS analyses since the results would supplement one another, e.g. elements below the detection limits of the XRF would be detected by the ICP-MS method."

3) Section 3.1 and 3.2 can be merged to a single section and the variability of trace metal composition in various sizes and total TM concentrations can be discussed.

We agree with Referee #2 to merge these two subsections into a single subsection. These two subsections were therefore combined into Section 3.1, which was renamed:

"3.1 Size-resolved concentrations and size distribution of trace metals"

All subsequent subsection numbers were changed accordingly in Section 3 (indicated with track changes).

Furthermore these two subsections were restructured as follows:

[revised manuscript text omitted]

4) Page 8, Line 7-9: How dust is impacting TM concentration? Its not clear. What is the source of dust? It is discussed by the authors that the sampling site is surrounded by pollutant emitting sources at least in the eastern region. However, there is no mention of dust source in the west or even eastern part of sampling site. Is there any hotspot for dust emission in the proximity of sampling site. Or is it local dust?

This discussion on the potential sources of wind-blown dust was expanded and restructured to form part of the last paragraph of Section 3.1 in the revised manuscript (combined Section 3.1 and 3.2 in the original manuscript). A new Figure 1 was compiled to include the major biomes for southern Africa, while 96-hour overlay back trajectories arriving hourly at Welgegund were compiled for the entire sampling period and included as supplementary material (Figure S1). In the discussion the arid Nama-Karoo biome was considered to be a potential regional source of dust, while agricultural activities were considered to be potential local sources. The following text was included in the last paragraph of Section 3.1 in the revised manuscript:

"From Figure 2 and 3 it is evident that a major source of trace metal species in all three size fractions can be considered to be wind-blown dust typically comprising Fe, Ca, Mg, Al, K and Ti (Polidori et al., 2009). As mentioned, Welgegund is a regional background location affected by air masses passing over

large pollutant source regions and a relatively clean background area (Figure 1). In Figure S1 96-hour overlay back trajectories arriving hourly at Welgegund for the entire sampling period (24 November 2010 until 28 December 2011) are presented. From Figure 1 and S1 it is evident that Welgegund is frequently impacted by long range transport of air masses passing over the relatively clean background region in the west (between north- and south-west). It is evident from Figure 1 that the arid Nama-Karoo biome is situated within this region west of Welgegund, which could be a potential regional source for wind-blown dust. In addition, Jaars et al., 2016 also indicated the extent of agricultural activities within a 60 km radius from Welgegund, which could be a significant local source of wind-blown dust. In addition, Figure S1 indicate that Welgegund is also frequently affected by air masses moving over the western Bushveld Igneous Complex, which is associated with a large number of pyrometallurgical smelters (e.g. ferrochrome, platinum and base metals) and mining activities (Venter et al, 2012, Tiitta et al., 2014; Jaars et al., 2014). This source region could therefore contribute to regional elevated levels of Fe, Cr, Ni, Zn, Mn and V measured at Welgegund. Venter at al., 2016 indicated that Cr(VI) concentrations were elevated in air masses that had passed over the western Bushveld Igneous Complex with the majority of Cr(VI) in the smaller $PM_{2.5}$ size fraction. The possible sources of trace metal species measured at Welgegund will be further explored in section 3.5."

5) Section 3.3 on comparing data set with previous studies from similar and other area is over discussed. Why air-quality aspect suddenly brought in the discussion. Does this study have any bearing on health issues?

A general comment made by Referee #3 specifically indicated that the results presented in this paper were adequately integrated with the existing body of knowledge. Therefore in view of this positive review of Referee #3 on the adequacy of the contextualisation of atmospheric trace metal concentration, Section 3.3 (3.2 in revised version) was considered by the authors to be adequately discussed. However, taking into consideration this comment by Referee #2, paragraphs 3-5 in Section 3.3, i.e. from Page 9 Line 27 – Page 11 Line 6, in the originally submitted manuscript (Section 3.2 in revised manuscript) were somewhat shortened and restructured as follows:

"Fe was also the most abundant species at Marikana and Rustenburg, with significantly higher concentrations compared to Welgegund. Mg was the second most abundant species at Marikana, while Mn and Cr concentrations were the second and third highest, respectively at Rustenburg. Cr levels at Rustenburg were approximately 2.5 times higher than levels thereof at Welgegund. However, Cr concentrations measured at Welgegund were approximately two times higher compared to Cr levels determined at Marikana, which could be attributed to the long range transport of Cr units (Figure 1 and S1). Venter et al., (2016) also indicated other combustion sources outside the western Bushveld Igneous Complex contributed to the atmospheric Cr(VI) concentrations at Welgegund. Ni and Zn concentrations at Welgegund were an order of magnitude lower compared to levels thereof at Marikana and

Rustenburg, while Mn and V concentrations were significantly lower compared levels thereof measured at Rustenburg. Similar to Welgegund, Na, B and Al were also relatively abundant at Marikana with concentrations of these species an order of magnitude higher at Marikana. Fe concentrations were similar at Vaal Triangle than levels thereof at Welgegund, while the annual average Na concentration was seven times higher and the annual average K level was an order of magnitude higher at the Vaal Triangle. Cr, Ni and Zn, typically associated with pyrometallurgical industries, were significantly lower in the Vaal Triangle compared to levels thereof at Welgegund. However, Mn concentrations at the Vaal Triangle were higher compared to levels thereof at Welgegund and Marikana. This can be attributed to the presence of a ferromanganese (FeMn) smelter in the Vaal Triangle region, as indicated in Figure 1.

The atmospheric trace metal concentrations determined at Welgegund were also compared to measurements at regional background sites near Beijing, China (Duan et al., 2012), the west coast of Portugal (Pio et al., 1996) and Spain (Querol et al. 2007). Al concentrations near Beijing were significantly higher compared to other trace metal species, while Na was the second most abundant species. Elevated levels of K, Fe and Ca were also determined near Beijing. Al, Na and K concentrations were an order of magnitude higher compared to levels of these species determined at Welgegund, while Fe levels were twice as low near Beijing All the other trace metal species measured near Beijing (with the exception of Ca, Pb and Mn) were an order or two orders of magnitude lower compared to concentrations of these species at Welgegund. Annual average trace metal concentrations determined at the two European regional background sites were an order or two orders of magnitude lower compared to trace metal levels determined at Welgegund. The generally lower trace metal concentration determined at these sites in China and Europe compared to Welgegund can be attributed to the sites in China and Europe being more removed from a conglomeration of metal sources."

Section 3.3 (3.2 in revised version) aims at contextualising the atmospheric trace metal concentrations measured at Welgegund, which include comparison to measurements in other parts in this region and in the world. Furthermore, the authors also considered comparison of atmospheric trace metal concentrations measured at Welgegund with air quality guidelines/standards as part of contextualising these trace metal concentrations. The aim was not to have any bearing on health issues, but rather to relate atmospheric trace metal concentrations measured at a regional background site to existing air quality guidelines/standards. Comparison of trace metal levels measured at Welgegund did indicate that Ni and As can be considered a regional problem that can be attributed to metal refining in the western Bushveld Igneous Complex. The heading of Section 3.3 (3.2 in revised version) was also changed as follows:

"3.2 Contextualisation of atmospheric trace metal concentrations"

6) Page 10, Line 4-5: Why and how dust can contribute Cr to the particulate matter?

This sentence was changed to indicate that the long range transport of Cr could be a source of Cr measured at Welgegund as indicated by the overlay back trajectories in Figure S1 and discussed in the last paragraph of Section 3.1 of the revised manuscript. In addition, the paper on Cr(VI) measurements at Welgegund also indicated that other combustion sources not within the western Bushveld Complex contributed to Cr(VI) concentrations at Welgegund. This discussion was changed as follows:

"…2.5 times higher than levels thereof at Welgegund. However, Cr concentrations measured at Welgegund were approximately two times higher compared to Cr levels determined at Marikana, which could be attributed to the long range transport of Cr units (Figure 1 and S1). Venter et al., (2016) also indicated other combustion sources outside the western Bushveld Igneous Complex contributed to the atmospheric Cr(VI) concentrations at Welgegund. Ni and Zn concentrations…"

7) Section 3.4: Seasonal trend cannot be discussed based on 1 yr data, however seasonal variability can be.

The heading of Section 3.4 (3.3 in revised version) was changed as follows:

"3.3 Seasonal variability

---

## Author Comment (AC2) · 13 Feb 2017

**General Comments:**

This is an easy to read and well written discussion paper on the analysis of trace metals in aerosol samples collected from a site in central South Africa. The extent and originality of the contribution to the understanding aerosol trace metal contributions are not new globally, but rather across the region, as is highlighted by the authors. The analytical methods employed are well established in literature. The figures/table are clear and evaluation of the generated results and their integration with the existing body of knowledge is sufficient. There is correct use of references and their presentation in a reference list.

The authors would like to thank Referee #3 for this very positive review on this paper and acknowledging the relevance of this work, especially, for this understudied region, i.e. southern Africa. As indicated in this paper only a few studies on atmospheric trace metal concentration have been conducted in South Africa that are published in peer-reviewed journals or available in the public domain. We also would like to thank Referee #3 for pointing out that the analytical methods employed in this study are well established in literature. Although there are deficiencies associated with the analytical method employed, i.e. unable to dissolve and extract silicate minerals, these analytical methods are commonly utilised for the analysis of atmospheric trace metals. This paucity was recognised and discussed by the authors (start of Section 1). We also thank Referee #3 for indicating that the results presented were adequately integrated with the existing body of knowledge.

Each of the scientific questions raised and technical corrections suggested by Referee #3 were addressed as indicated below in order to further improve the scientific content of the manuscript. All changes in the manuscript are indicated with track changes.

**Scientific Questions:**

Pg 8 Line 24 to 27. This statement should be explored further in this study and supported by clear justification based on the understanding of long range transportation of pollution in the region. Is this statement supported by e.g., trajectory analysis of air masses to Welgegund?

We thank Referee #2 for pointing out the significance of indicating the long range transport of pollution influencing this region. In an effort to address this comment 96-hour overlay back trajectories arriving

hourly at Welgegund were compiled for the entire sampling period and included as supplementary material. From this back trajectory analysis the influence of air masses passing over the major pollutant source regions is evident, especially, with regard to pyrometallurgical smelters and mining activities located in the western Bushveld Igneous Complex. Referee #2 also requested justification of the possible long-range and local sources of wind-blown dust. Furthermore, Section 3.1 and 3.2 was combined and restructured according to a suggestion made by Referee #2. The following text was included in the last paragraph of Section 3.1 in the revised manuscript:

"From Figure 2 and 3 it is evident that a major source of trace metal species in all three size fractions can be considered to be wind-blown dust typically comprising Fe, Ca, Mg, Al, K and Ti (Polidori et al., 2009). As mentioned, Welgegund is a regional background location affected by air masses passing over large pollutant source regions and a relatively clean background area (Figure 1). In Figure S1 96-hour overlay back trajectories arriving hourly at Welgegund for the entire sampling period (24 November 2010 until 28 December 2011) are presented. From Figure 1 and S1 it is evident that Welgegund is frequently impacted by long range transport of air masses passing over the relatively clean background region in the west (between north- and south-west). It is evident from Figure 1 that the arid Nama-Karoo biome is situated within this region west of Welgegund, which could be a potential regional source for wind-blown dust. In addition, Jaars et al., 2016 also indicated the extent of agricultural activities within a 60 km radius from Welgegund, which could be a significant local source of wind-blown dust. In addition, Figure S1 indicate that Welgegund is also frequently affected by air masses moving over the western Bushveld Igneous Complex, which is associated with a large number of pyrometallurgical smelters (e.g. ferrochrome, platinum and base metals) and mining activities (Venter et al, 2012, Tiitta et al., 2014; Jaars et al., 2014). This source region could therefore contribute to regional elevated levels of Fe, Cr, Ni, Zn, Mn and V measured at Welgegund. Venter at al., 2016 indicated that Cr(VI) concentrations were elevated in air masses that had passed over the western Bushveld Igneous Complex with the majority of Cr(VI) in the smaller $PM_{2.5}$ size fraction. The possible sources of trace metal species measured at Welgegund will be further explored in section 3.5."

Pg 11 Line 14. Throughout the manuscript, the phrase 'standard limit' has been used. Please note that a STANDARD is the limit value. Therefore to avoid redundancy, use STANDARD without limit.

The phrase "standard limit" was replaced with the term "standard" throughout the document.

Pg 11 Section 3.5. For the average concentrations, where are the standard deviations and how significant are they? How would these deviations affect the PCFA receptor modelling results, if at all they do?

For the PCFA all the concentrations determined with ICP-MS for each of the metal species in the PM1 size fraction were included, i.e. the concentration of each metal species determined for each PM1

sample collected was included. The instrument took three readings for each concentration point from which the relative standard deviation (%RSD) was calculated. The %RSD for all the metals analysed with ICP-MS ranged between approximately 0.2 and 5%, with the %RSP generally being below 2%. Therefore, these low %RSDs are not considered to influence PCFA results reported.

**Technical Corrections:**

Pg 2, Line 27 should be referenced IPCC, 2014.

"IPCC 2014" was changed to "IPCC, 2014" in the text.

Consistency in the use of AND or & in references siting

We thank Referee #3 for pointing this out. "&" was replaced with "and" in the reference citations and -list throughout the manuscript.

Pg 3, Line 28 should be referenced WHO, 2005

"WHO 2005" was changed to "(WHO, 2005)" in the text.

Pg 4 Lines 4, 5 and 6 and Pg 11 Line 11 "Government Gazette" should be defined based on author DEAT or DEA depending on the years.

"Government Gazette" was defined based on author "DEAT" and "DEA" in the text as follows:

"…regions being classified through legislation as air pollution priority areas, i.e. Vaal Triangle Airshed Priority Area (DEAT, 2006), Highveld Priority Area (DEAT, 2007) and Waterberg-Bojanala Priority Area (DEA, 2012). Air quality outside…" and "…the European Commission Air Quality Standards (ECAQ, 2008) and the South African National Air Quality Standards of the South African Department of Environmental Affairs (DEA) (DEA, 2009). There are currently only…"

The references in the "Reference" list was also changed accordingly:

"DEA, Department of Environmental Affairs. 2009. National Environmental Management: Air Quality Act, 2004 (ACT NO. 39 OF 2004) National ambient air quality standards, Government Gazette, 24 December 2009, pp. 6-9.

DEA, Department of Environmental Affairs. 2012. Notice 495 of 2012. Department of Home Affairs, National Environmental Management: Air Quality Act, 2004, Declaration of the Waterberg National Priority Area, South African Government Gazette No. 35345 on 15 June 2012; Correction notice (154):

Waterberg-Bojanala National Priority Area, South African Government Gazette No. 36207 on 8 March 2013."

DEAT, Department of Environmental Affairs and Tourism. 2006. Declaration of the Vaal Triangle Airshed Priority Area in terms of section 18(1) of the National Environmental Management: Air Quality Act 2004 (Act no. 39 of 2004), Government Gazette, 21 April 2006.

DEAT, Department of Environmental Affairs and Tourism. 2007. Department of Environmental Affairs and Tourism. Declaration of the Highveld as priority area in terms of section 18(1) of the National Environmental Management: Air Quality Act 2004 (Act no. 39 of 2004), Government gazette, 23 November 2007.

Pg 5 Line 11, Use full text on first mention e.g., Desulfurization (DeSOx) / Denitrification (DeNOx) equipment.

The text was changed as follows:

"…with other large point sources, such as two coal-fired power stations (without desulphurisation (de-SOx) and denitrification (de-NOx)) and large pyrometallurgical…"

Pg 5 Line 26, Expand ICP-MS on first mention.

ICP-MS was written out in full and abbreviated on Pg 6 line 3 (Section 2.2) (in the original manuscript submitted) where it was first mentioned in the text.

Pg 11 Line 23 Should this be 2.5x10E4 or 10E-4? Please check Table 1 as well.

Thank you for Referee #2 for pointing this out. This value was checked and the correct value is 2.5x10E-5, which was changed in the text and in Table 1.

For references, check for consistency in the use of DOI/doi throughout the discussion paper.

"DOI" was changed to "doi" throughout the Reference list.